# Efficient non-Markovian quantum dynamics using time-evolving matrix product operators

A. Strathearn[1], P. Kirton [1], D. Kilda[1], J. Keeling [1] & B.W. Lovett [1]

In order to model realistic quantum devices it is necessary to simulate quantum systems strongly coupled to their environment. To date, most understanding of open quantum systems is restricted either to weak system–bath couplings or to special cases where specific numerical techniques become effective. Here we present a general and yet exact numerical approach that efficiently describes the time evolution of a quantum system coupled to a non-Markovian harmonic environment. Our method relies on expressing the system state and its propagator as a matrix product state and operator, respectively, and using a singular value decomposition to compress the description of the state as time evolves. We demonstrate the power and flexibility of our approach by numerically identifying the localisation transition of the Ohmic spin-boson model, and considering a model with widely separated environmental timescales arising for a pair of spins embedded in a common environment.

---

[1] SUPA, School of Physics and Astronomy, University of St Andrews, St Andrews KY16 9SS, UK. These authors contributed equally: A. Strathearn, P. Kirton. Correspondence and requests for materials should be addressed to B.W.L. (email: bwl4@st-andrews.ac.uk)

The theory of open quantum systems describes the influence of an environment on the dynamics of a quantum system[1]. It was first developed for quantum optical systems[2], where the coupling between system and environment is weak and unstructured. In such situations, one can almost always assume that the environment is memoryless and uncorrelated with the system—that is, the Markov and Born approximations hold—allowing a time-local equation of motion to be derived for the open system. The resulting Born–Markov master equation works because the environment-induced changes to the system dynamics are slow relative to the typical correlation time of the environment.

There are now a growing number of quantum systems where a structureless environment description is not justified, and memory effects[3] play a significant role. These include micromechanical resonators[4], quantum dots[5,6] and superconducting qubits[7], and can underpin emerging quantum technologies such as the single-photon sources needed for quantum communication[8]. In addition, structured environments are ubiquitous in problems involving the strong interplay of vibrational and electronic states. For example, those involving the photophysics of natural photosynthetic systems[9,10], complex organic molecules used for light emission or solar cells[11], or semiconductor quantum dots[12–15]. Similar problems arise when considering non-equilibrium energy transport in molecular systems[16] or non-adiabatic processes in physical chemistry[17]. Non-Markovian effects can even be a resource for quantum information[18,19].

Various approaches exist for dealing with non-Markovian dynamics[1,3]. Some particular problems have exact solutions[20]. For others, unitary transformations can uncover effective weak coupling theories, and perturbative expansions beyond the Born–Markov approximations[12,21]; these techniques typically yield time-local equations and are limited to certain parameter regimes. Diagrammatic formulations of such perturbative expansions can also form the basis for numerically exact approaches, for example, the real-time diagrammatic Monte Carlo as implemented in the Inchworm algorithm[22,23]. Finally, there are non-perturbative methods that enlarge the state space of the system. This can be through hierarchical equations of motion[24], through capturing part of the environment within the system Hilbert space[25–27] or by using augmented density tensors (ADTs) to capture the system's history[28,29]. These can be very powerful but require either specific assumptions about the environments[24,27] or resources that scale poorly with bath memory time.

In this Article, we describe a computationally efficient, general and yet numerically exact approach to modelling non-Markovian dynamics for an open quantum system coupled to an harmonic bath. Our method, which we call the time-evolving matrix product operator (TEMPO), exploits the ADT[28,29] to represent a system's history over a finite bath memory time $\tau_c$. If the bath is well behaved, then using a singular value decomposition (SVD) to compress the ADT on the fly is expected to enable accurate calculations with computational resources scaling only polynomially with $\tau_c$. We demonstrate the power of TEMPO by exploring two contrasting problems: the localisation transition in the spin-boson model (SBM)[30] and spin dynamics with an environment that has both fast and slow correlation timescales—a problem for which other methods are not available. For both these problems we observe polynomial scaling with memory time.

## Results

**Time-evolving matrix product operators**. In this section, we outline how the TEMPO algorithm works; further details are provided in the Methods section. We start by introducing the ADT. To define the notation and our graphical representation of it, we first consider the evolution of a Markovian system, which can be described by a density operator that contains $d^2$ numbers for a $d$-dimensional Hilbert space. Usually, the density operator is written as a $d \times d$ matrix, but we instead use a length $d^2$ vector with elements $\rho^i(t)$. To evolve by a timestep $\Delta$, we write

$$\rho^i(t + \Delta) = \left[ e^{\Delta \mathcal{L}} \right]^i_j \rho^j(t), \tag{1}$$

where $\mathcal{L}$ is the Liouvillian[1]. The graphical representation of this is shown in Fig. 1a. The red circle represents the density operator, with the protruding 'leg' indicating this is a tensor of rank one, that is, a vector. This leg is indexed by an integer $i = 1, \ldots, d^2$. The blue square with two legs represents the propagator $e^{\Delta \mathcal{L}}$, written as a $d^2 \times d^2$ superoperator[1]. The matrix–vector multiplication in Eq. (1) is shown by joining a leg of the propagator to the density operator, indicating tensor contraction. This contraction generates the density operator at time $t + \Delta$.

In order to capture non-Markovian dynamics, we extend our representation of the state at time $t$ from a vector to an ADT, representing the history of the system. This is motivated by the path integral of a system interacting linearly with a bosonic environment. After integrating out the environment, the influence of the environment on the system can be captured by an 'influence functional' of the system paths alone[1]. The influence functional couples the current evolution to the history, and captures the non-

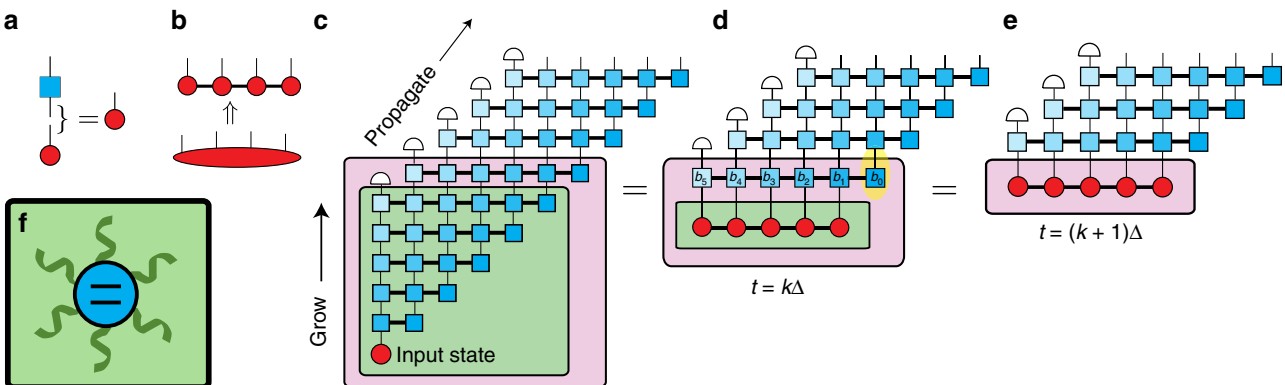

**Fig. 1** Schematic description of the TEMPO algorithm. **a** Pictorial representation of matrix–vector multiplication. In **b** we show how the ADT can be decomposed into an MPS. **c** The full tensor network starting from an initial standard density operator which is grown to an ADT with $K$ legs, as shown in **d**, where we have contracted the contents of the green box. To propagate forward one step, we contract the ADT with the next row of the propagator, as in **e**. A schematic representation of the spin-boson model is shown in **f**

Markovian dynamics. Makri and Makarov[28,29] showed that by considering discrete timesteps, and writing the sum over system states in a discrete basis, the path integral could be reformulated as a propagator for the ADT, written as a discrete *sum* over paths. The influence functional becomes a series of influence functions $I_k(j, j')$ that connect the evolution of the amplitude of state $j$ to the amplitudes of states $j'$ an integer number, $k$, of timesteps ago. This approach is known as the quasi-adiabatic path integral (QUAPI).

As described so far, the ADT grows at each timestep, to record the lengthening system history. However, the influence functions have no effect once $k\Delta$ exceeds the bath correlation time $\tau_c$. One can therefore propagate an ADT containing only the previous $K = \tau_c/\Delta$ steps: this is the finite memory approximation. This means we consider an ADT of rank $K$, written as $A^{i_1, i_2, \cdots, i_K}(t)$, where each index runs over $i_k = 1, \ldots, d^2$. The explicit construction of this tensor is described in the Methods section. In general $A^{i_1, i_2, \cdots, i_K}(t)$ contains $d^{2K}$ numbers, which scales exponentially with the correlation time $\tau_c$. If the full tensor is kept, one quickly encounters memory problems, and typical simulations are restricted to $K < 20$[31,32]. Improved QUAPI algorithms[33,34] show that (for some models) typical evolution does not explore this entire space, leading us to seek a minimal representation of the ADT.

Matrix product states (MPS)[35,36] are natural tools to represent high-rank tensors efficiently where correlations are constrained in some way. Examples include the ground state of one-dimensional (1D) quantum systems with local interactions[37], steady state transport in 1D classical systems[38] or time-evolving 1D quantum states[39]. Inspired by these results, we show how an ADT can be efficiently represented and propagated using standard MPS methods. One may decompose high-rank tensors into products of low-rank tensors using SVDs and truncation. By combining indices, the tensor $A$ can be written as[36]:

$$A_{\{i_1, \ldots, i_k\}, \{i_{k+1}, \ldots, i_K\}} = U_{\{i_1, \ldots, i_k\}, \alpha} \lambda_\alpha \left[V^\dagger\right]_{\alpha, \{i_{k+1}, \ldots, i_K\}}. \quad (2)$$

Here, $U$, $V$ are unitary matrices, and $\lambda_\alpha$ denotes a singular value of the matrix $A$. Truncation corresponds to throwing away singular values $\lambda_\alpha$ smaller than some cutoff $\lambda_c$, consequently reducing the size of the matrices $U$, $V$. This procedure can be iterated by sweeping $k$ across the whole tensor. The result of this is shown graphically in Fig. 1b, and can be written as:

$$A^{i_1, \ldots, i_k, \ldots, i_K} = a^{i_1}_{\alpha_1} a^{i_2}_{\alpha_1, \alpha_2} \cdots a^{i_k}_{\alpha_{k-1}, \alpha_k} \cdots a^{i_K}_{\alpha_{K-1}}. \quad (3)$$

This provides an efficient representation of the state, with a precision controlled by $\lambda_c$.

$A^{i_1, i_2, \cdots, i_K}(t)$ can be time locally propagated using a tensor $B^{j_1, \cdots, j_K}_{i_1, \cdots, i_K}$. Crucially, this propagation can be performed directly on the matrix product representation of $A$. Moreover, the tensor product description of $B^{j_1, \cdots, j_K}_{i_1, \cdots, i_K}$, shown as the connected blue squares in Fig. 1c, has a small dimension, $d^2$, for the internal legs. Similarly to the time evolution shown in Fig. 1a, the state $A(t + \Delta)$ is generated by contracting the legs of $A(t)$ with the input legs of $B$. Contracting a tensor network with a MPS, and truncating the resulting object by SVDs is a standard operation[36]. In all the applications we discuss below, we find that as time propagates we are able to maintain an efficient representation of $A^{i_1, i_2, \cdots, i_K}(t)$ with precision determined by $\lambda_c$.

The structure of the propagator depends on the influence functions $I_k(j, j')$ as shown in Fig. 1c (see also Methods section). We use darker colours to represent influence functions corresponding to more recent time points, which are expected to generate stronger correlations in the ADT. The input and output legs of the propagator are offset in the figure, so time can be viewed as propagating from left to right. In effect, at each step

the register is shifted so that the right-most output index corresponds to the new state: events that occurred more than $\tau_c$ ago are dropped, as illustrated by the white semicircles in Fig. 1, since they do not influence the future evolution. Evolution over a series of timesteps is depicted in Fig. 1c–e. In Fig. 1c we show the full tensor network. Assuming the initial state of the system is uncorrelated with its environment means it can be drawn as a regular density operator. In the 'grow' phase, a series of asymmetric B propagators are applied, which allow the relevant system correlations to extend in time. Once the system has grown to an object with $K$ legs, we enter the regular propagation phase, shown in Fig. 1d, e.

**Spin-boson phase transition**. To demonstrate the utility of the TEMPO algorithm, we apply it to two problems of a quantum system coupled to a non-Markovian environment. We first consider the unbiased SBM[30], which has long served as the proving ground for open system methods. The generic Hamiltonian of this model is

$$H = \Omega S_x + \sum_i S_z \left(g_i a_i + g_i^* a_i^\dagger\right) + \omega_i a_i^\dagger a_i, \quad (4)$$

where the $S_i$ are the usual spin operators, $a_i^\dagger (a_i)$ and $\omega_i$ are, respectively, the creation (annihilation) operators and frequencies of the $i$th bath mode, which couples to the system with strength $g_i$. The behaviour of the bath is characterised by the spectral density function

$$J(\omega) = \sum_i |g_i|^2 \delta(\omega - \omega_i). \quad (5)$$

This model is known to show a rich variety of physics depending on the particular form of spectral density and system parameters chosen. When the spectral density is Ohmic, $J(\omega) = 2\alpha\omega \exp(-\omega/\omega_c)$, the model is known to exhibit a quantum phase transition in the BKT universality class[40], at a critical value of the system–environment coupling $\alpha = \alpha_c$[30,41]. The transition takes the system from a delocalised phase below $\alpha_c$, where any spin excitation decays ($\langle S_z \rangle = 0$ in the steady state), to a localised phase above $\alpha_c$ ($\langle S_z \rangle \neq 0$ in the steady state). Most analytic results are restricted to the regime where the cutoff frequency $\omega_c \gg \Omega$. For example, when $S$ describes a spin-1/2 particle, the phase transition occurs at $\alpha_c = 1 + \mathcal{O}(\Omega/\omega_c)$[30,40,42].

We are able to explore the dynamics around this phase transition using TEMPO. In Fig. 2a we show the polarisation dynamics of the spin-1/2 SBM for a range of $\alpha$ at $K = 200$. This memory length is an order of magnitude larger than standard ADT implementations[30] and is required to reach the asymptotic limit of the dynamics in the vicinity of the phase transition. We achieve convergence by varying the timestep $\Delta$ and SVD cutoff $\lambda_c$. We take an initial condition $\langle S_z \rangle = +1/2$ with no excitations in the environment, and find $\langle S_z(t) \rangle$.

Before reaching the localisation transition at $\alpha = \alpha_c$, one first reaches a crossover at $\alpha \simeq 0.5$ from coherent decaying oscillations to incoherent decay[29]. For $\alpha > 0.5$, we find $\langle S_z \rangle$ always decays to zero asymptotically as $\langle S_z(t) \rangle \propto \exp(-\gamma t)$ to a very good approximation; fits to this function are shown as dashed lines in Fig. 2a. Decay to zero for all $\alpha > 0.5$ conflicts with the existence of a localised phase at large $\alpha$, where $\langle S_z \rangle$ should asymptotically approach a non-zero value. The origin of this discrepancy is the finite memory approximation, which produced a time-local equation in the enlarged space of $K$ timesteps. Time-local dynamics of a finite system typically generates a gapped spectrum of the effective Liouvillian[43]. In the localised phase, $\alpha > \alpha_c$, the spectral gap should vanish asymptotically as we increase

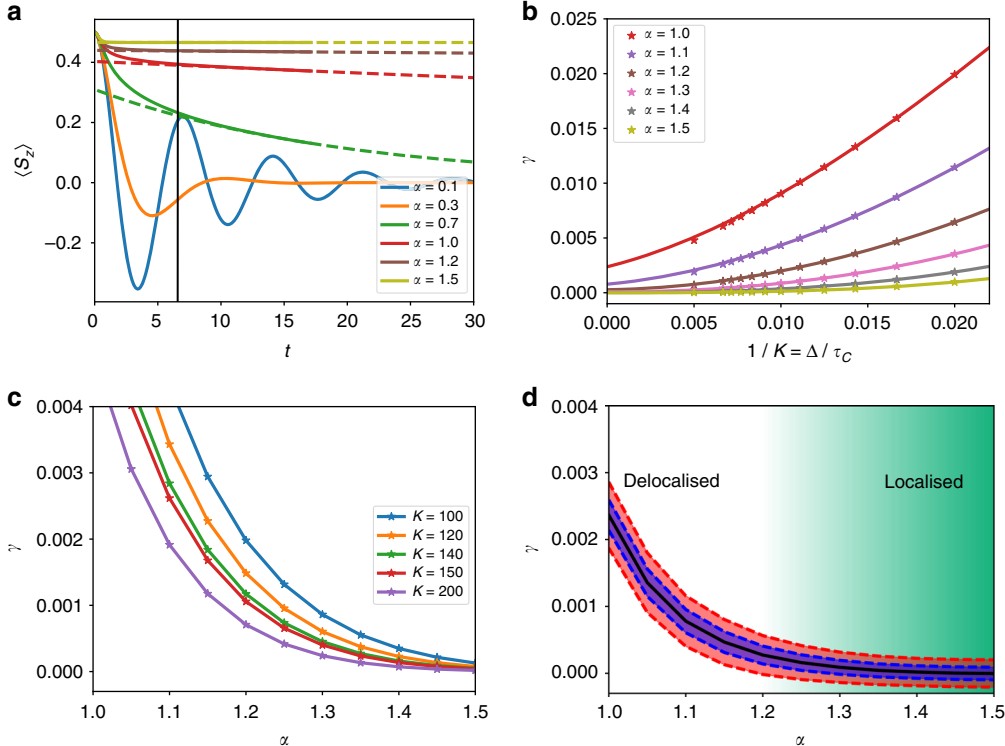

**Fig. 2** Behaviour of the spin-1/2 system through the localisation phase transition. **a** The dynamics captured at $K = 200$ for the values of $\alpha$ indicated. The dotted lines show the exponential fits to the data above $\alpha = 0.5$. The vertical black line shows the location of the memory cutoff used. **b** The dependence of the decay rate of the exponential fit on $1/K$. This allows us to analyse the behaviour as $K \to \infty$. **c** The change in the decay rate as we go through the transition by varying $\alpha$ for the values of $K$ indicated. In **d**, we give 68% (blue) and 95% (red) confidence intervals for the extrapolated decay rate which crosses zero at around $\alpha_c \simeq 1.25$. The bath cutoff frequency is $\omega_c = 5$ and everything is measured in units of the Hamiltonian driving term $\Omega$

the memory cutoff $\tau_c \to K\Delta$. We should thus examine how the extracted decay rate, $\gamma$, depends on the memory cutoff. For $\alpha < \alpha_c$, $\gamma$ should remain finite as $\tau_c \to \infty$, while for $\alpha > \alpha_c$ it should vanish. In Fig. 2b we plot $\gamma$ as a function of $1/K = \Delta/\tau_c$ for different values of $\alpha$ around the phase transition. At small $\alpha$, $\gamma$ does appear to remain finite as $K \to \infty$, while at large $\alpha$ the behaviour appears consistent with localisation.

We may estimate the location of the phase transition by extrapolating $1/K \to 0$ for each $\alpha$, and finding the smallest value of $\alpha$ consistent with $\gamma \to 0$. To do this, we use cubic fits in Fig. 2b (solid lines), and extract the constant part, with the restriction that the extracted $\gamma$ cannot be negative. In order to find the phase transition as accurately as possible, we must perform simulations up to very large values of $K$: we here perform simulations up to $K = 200$, something that would be simply impossible without the tensor compression we exploit. Errors in our fits are assessed by monitoring the sensitivity of the best-fit result to truncation precision $\lambda_c$. These errors are all $<10^{-4}$ and so are smaller than the points in Fig. 2. This allows us to find an error in the extracted $K \to \infty$ limit. The extracted values for $\gamma$ are displayed in Fig. 2d where we show our estimate for its 68 and 95% confidence intervals. These suggest that $\alpha_c \simeq 1.25$, consistent with the known analytic results[30,40,42]. We note that identifying $\alpha_c$ precisely from the time dependence of $\langle S_z \rangle$ is particularly challenging: since the localisation transition is in the BKT class[40], the order parameter approaches zero continuously.

The efficiency of TEMPO enables consideration of models with a larger local Hilbert space. To demonstrate this, we examine the localisation transition in the spin-1 SBM. Physically this could either arise from a spin-1 impurity or from a pair of spin-1/2 particles interacting with a common environment[44]. On switching to this problem, the local dimension of each leg of our state

tensor increases from $d^2 = 4$ to $d^2 = 9$, reducing the values of $K$ we can reach. However, we also find convergence occurs for larger timesteps, allowing access to similar values of $\tau_c$.

In Fig. 3a we show the dynamics of this model, after initialising to $\langle S_z \rangle = 1$. In this case, on both sides of the localisation transition, the dynamics shows complex oscillatory behaviour before settling down to an exponential decay. This introduces more uncertainty to our exponential fits. However, as shown in Fig. 3b the extracted decay rate vanishes at $\alpha_c \simeq 0.28$, indicative of the phase transition and agreeing with numerical renormalisation group results[44,45], but in contrast to the results found using a variational ansatz[46].

**Two spins in a common environment**. We next demonstrate the flexibility of TEMPO by applying it to a dynamical problem for which other methods are not available. We consider a pair of identical spins-1/2, at positions $\mathbf{r}_a$ and $\mathbf{r}_b$, which couple directly to each other through an isotropic Heisenberg coupling $\Omega$, and which both couple to a common environment, see Fig. 4a. The Hamiltonian reads:

$$H = \Omega \mathbf{S}_a \cdot \mathbf{S}_b + \sum_{\nu=a,b} \sum_i S_{z,\nu} \left( g_{i,\nu} a_i + g_{i,\nu}^* a_i^\dagger \right) + \omega_i a_i^\dagger a_i. \quad (6)$$

The system–bath coupling constants have a position-dependent phase, $g_{i,\nu} = g_i e^{-i\mathbf{k}_i \cdot \mathbf{r}_\nu}$, where $\mathbf{k}_i$ is the wavevector of the $i$th bosonic mode. We assume linear dispersion $\omega_i = c|\mathbf{k}_i|$ and $c = 1$.

This model exhibits complex dissipative dynamics on two different timescales. The faster timescale describes dissipative dynamics of the spins due to interactions with their nearby environment, typically set by the $\omega_c$ defined earlier. The other timescale is set by the spin separation $R = |\mathbf{r}_a - \mathbf{r}_b|$ over which there is an environment-mediated spin–spin interaction. By

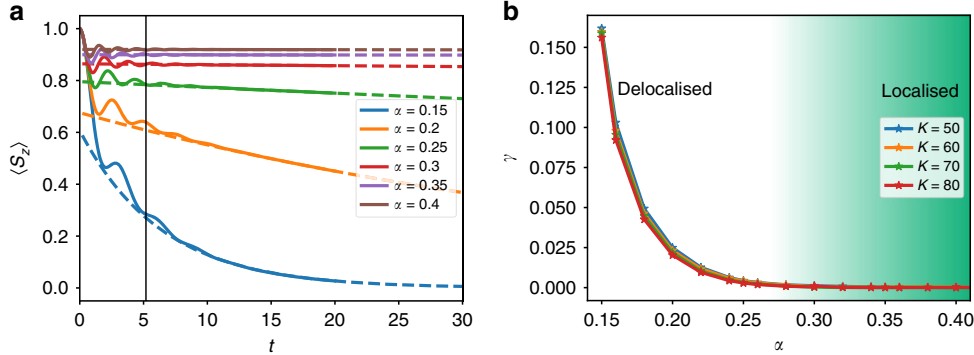

**Fig. 3** Behaviour of the spin-1 system through the localisation phase transition. **a** The dynamics captured at $K = 80$ for the values of $\alpha$ indicated. The dotted lines show the exponential fits to the data. **b** The change in the decay rate as we go through the transition by varying $\alpha$ for the values of $K$ indicated. The system parameters are the same as Fig. 2

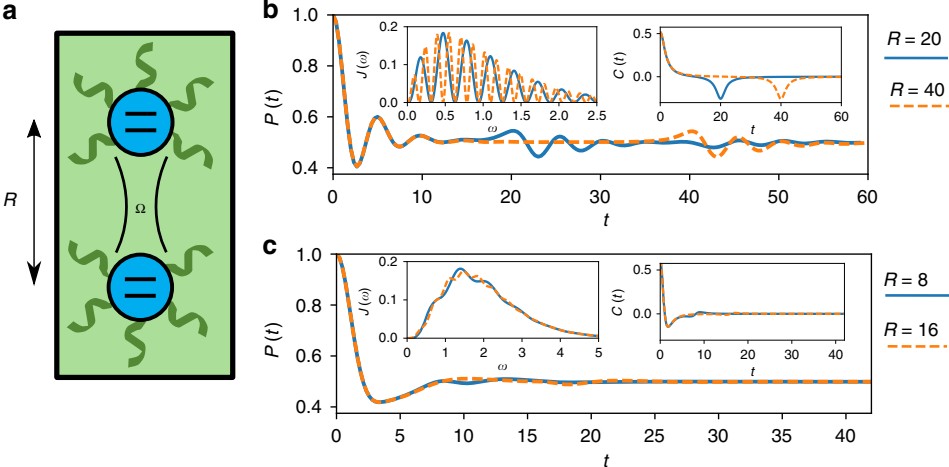

**Fig. 4** Dynamics of two coupled spins-1/2, separated by a distance $R$, interacting with the same environment. **a** A schematic of this system. **b**, **c** Dynamics of the system in 1D and 3D, respectively, at different values of the spin separation $R$. Insets to these plots are the corresponding spectral densities and bath correlation functions (see Methods section for details). The dimensionless couplings $\alpha$ used for 1D and 3D are $\alpha = 2$ and $\alpha = 1$, respectively. We set the speed of sound $c = 1$, so that all parameters are in units of $\Omega$ and we choose $T = 0.5$, $\omega_c = 0.5$. In all cases we have used 180 timesteps, but not used the memory cutoff meaning $K = 180$

changing $R$ we can control the ratio of these timescales. The dimension, $D$, of the bath also has an effect: the intensity of environmental excitations propagating from one spin to the other will be stronger for lower $D$.

When the spins are close together, $R < \omega_c^{-1}$, it is difficult to distinguish local dissipative effects from the environment-mediated interaction and both master equation techniques[13] and the standard ADT method[47] generate accurate dynamics. Instead, we consider large separation $R > \omega_c^{-1}$, about which little is known. The ADT then requires both a small timestep $\Delta \ll \omega_c^{-1}$ to capture the fast local dissipative dynamics and a large cutoff time $\tau_c = K\Delta > R$ to capture environment-induced interactions; hence, a very large $K$ is needed. Using TEMPO we are able to investigate these dynamics without even having to go beyond the tensor growth stage shown in Fig. 1c, and thus avoid any error caused by a finite memory cutoff $K$.

We project onto the $S_{z,a} + S_{z,b} = 0$ subspace of the system, consisting of the two anti-aligned spin states, since this is the only sector with non-trivial dynamics. The effective Hamiltonian for this $2d$ subspace can then be mapped onto the spin-1/2 SBM, Eq. (4), albeit with a modified spectral density that depends on $R$. Details of this procedure are given in the Methods section.

In Fig. 4b, c we show dynamics for different $R$ for environments with $D = 1$ and $D = 3$. Insets show the effective spectral densities, $J(\omega)$, and real part of the bath autocorrelation functions, $C(t)$, which we define in the Methods section. We initialise the spins in a product state with $\langle S_{z,a} \rangle = 1/2$, $\langle S_{z,b} \rangle = -1/2$ and calculate the probability, $P(t)$, of finding the system in this state at time $t$. The bath is initialised in thermal equilibrium at temperature $T$. For $D = 1$, after initial oscillations decay away over a timescale $\sim \omega_c^{-1}$, there are revivals at $t = R$. This is due to the strongly oscillating spectral density which results in a large peak at $C(t = R)$. As expected for a one-dimensional environment, the profile of these secondary oscillations is independent of $R$ when $R \gg \omega_c^{-1}$. Additionally for $R = 20$ more small amplitude oscillations appear at $t \approx 40$, due to the effective interaction of the spins at $t \approx 20$ sending more propagating excitations into the environment. For $D = 3$ the spectral density still has an oscillatory component, though it is much less prominent. The resulting peaks at $C(t = R)$ are thus much smaller than the $t = 0$ peak and have only a small effect on the dynamics. Small amplitude oscillations can be seen at $t \approx R$ when $R = 8$, but with $R = 16$ it is difficult to see any significant features in the dynamics.

## Discussion

We have presented a highly efficient method for modelling the non-Markovian dynamics of open quantum systems. Our method is applicable to a wide variety of situations. In well-established ADT methods, non-Markovianity is accounted for by encoding the system's history in a high-rank tensor; we have overcome the restrictive memory requirements of storing this tensor by representing it as an MPS. We can then efficiently calculate open system dynamics by propagating this MPS via iterative application of an MPO. To test our technique we used it to find the localisation transition in the SBM, for both spin-1/2 and spin-1, and found estimates for the critical couplings, consistent with other techniques. We then applied our method to a pair of interacting spins embedded within a common environment, in a regime where a large separation of timescales prevents the use of other methods.

Precisely locating the phase transition is a rigorous test of any numerical method: as we found, very large memory times, up to $K = 200$ were required to precisely locate this point. Other improved numerical methods[22,23,33] have demonstrated a degree of enhanced efficiency when considering conditions away from the critical coupling. As yet, other such general methods have not been used to precisely locate the transition.

The key to our technique is that tensor networks provide an efficient representation of high-dimensional tensors encoding restricted correlations. As well as the widespread application of such methods in low-dimensional quantum systems[35–39], they have also been applied to sampling problems in classical statistical physics[48], and analogous techniques (under the name 'Tensor trains') have been developed in computer science[49]. Moreover, there has been a recent synthesis showing how techniques developed in one context can be extended to others, such as machine learning[50], or Monte Carlo sampling of quantum states[51]. Our work defines a further application for these methods, and future work may yet yield even more efficient approaches.

The methods described in this article are already very powerful in their ability to model general non-Markovian environments. They also enable easy extension to study larger quantum systems, by adapting other methods from tensor networks such as the optimal boson basis[52]—these will be the subject of future work. They may also be combined with approaches such as the tensor transfer method described in Ref. [53]. This method allows efficient long time propagation of dynamics, so long as an exact map is known up to the bath memory time: TEMPO enables efficient calculation of the required exact map. With such tools available, the study of the dynamics of quantum systems in non-Markovian environments[3] can now move from studying isolated examples to elucidating general physical principles, and modelling real systems.

## Methods

**TEMPO algorithm**. In this section, we will present the details of the TEMPO algorithm, paying particular attention to how the ADT and propagator are constructed in a matrix product form.

The generic Hamiltonian of the models we consider is

$$H = H_0 + O \sum_i \left( g_i a_i + g_i^* a_i^\dagger \right) + \sum_i \omega_i a_i^\dagger a_i, \tag{7}$$

$$= H_0 + H_E, \tag{8}$$

where $H_0$ is the (arbitrary) free system Hamiltonian and $H_E$ contains both the bath Hamiltonian and the system–bath interaction. Here $a_i^\dagger$ ($a_i$) and $\omega_i$ are the creation (annihilation) operators and frequencies of the $i$th environment mode. The system operator $O$ couples to bath mode $i$ with coupling constant $g_i$. As outlined in the main text, we work in a representation where $d \times d$ density operators are given

instead by vectors with $d^2$ elements. These vectors are then propagated using a Liouvillian as in Eq. (1) of the main text, $\mathcal{L} = \mathcal{L}_0 + \mathcal{L}_E$, where $\mathcal{L}_0$ and $\mathcal{L}_E$ generate coherent evolution caused by $H_0$ and $H_E$, respectively. It has been shown recently that it is straightforward to include additional Markovian dynamics in the reduced system Liouvillian[54] in the ADT description.

If the total propagation over time $t_N$ is composed of $N$ short time propagators $e^{t_N \mathcal{L}} = (e^{\Delta \mathcal{L}})^N$, we can use a Trotter splitting[55]

$$e^{\Delta \mathcal{L}} \approx e^{\Delta \mathcal{L}_E} e^{\Delta \mathcal{L}_0} + \mathcal{O}(\Delta^2). \tag{9}$$

We note that the following arguments can be easily adapted to use the higher-order, symmetrized, Trotter splitting[28,29,56] that reduces the error to $\Delta^3$. All the numerical results presented use this symmetrized splitting, but for ease of exposition we use the form of Eq. (9) here. We assume the initial density operator factorises into system and environment terms, with the environment initially in thermal equilibrium at temperature $T$. Time evolution can then be written as a path sum over system states, by inserting resolutions of identity between each $e^{\Delta \mathcal{L}_E} e^{\Delta \mathcal{L}_0}$ and then tracing over environmental degrees of freedom. The result is the discretized Feynman–Vernon influence functional[28,29], which yields the following form for the time evolved density matrix:

$$\rho_{j_N}(t_N) = \sum_{j_1, \dots, j_{N-1}} \left( \prod_{n=1}^N \prod_{k=0}^{n-1} I_k(j_n, j_{n-k}) \right) \rho_{j_1}(\Delta). \tag{10}$$

The indexing here is in a basis where $O$ is diagonal. Each $j$ index runs from 1 to $d^2$, and due to the order of the splitting in Eq. (9), the initial state of the system has been propagated forward a single timestep, $\rho_{j_1}(\Delta) = \left[ e^{\Delta \mathcal{L}_0} \right]_{j_1 j_0} \rho_{j_0}(0)$. We have defined the influence functions

$$I_k(j, j') = \begin{cases} e^{\phi_k(j,j')}, & k \neq 1, \\ \left[ e^{\Delta \mathcal{L}_0} \right]_{jj} e^{\phi_1(j,j')}, & k = 1, \end{cases} \tag{11}$$

with

$$\phi_k(j, j') = -O_j^- \left( O_{j'}^- \mathrm{Re}[\eta_k] + i O_{j'}^+ \mathrm{Im}[\eta_k] \right). \tag{12}$$

Here $O_j^-$ are the $d^2$ possible differences that can be taken between two eigenvalues of $O$ and $O_j^+$ the corresponding sums. The coefficients, $\eta_k$, quantify the non-Markovian correlations in the reduced system across $k$ timesteps of evolution and are given by the integrals

$$\eta_{n-n'} = \begin{cases} \int_{t_{n-1}}^{t_n} dt' \int_{t_{n'-1}}^{t_{n'}} dt'' C(t' - t''), & n \neq n', \\ \int_{t_{n-1}}^{t_n} dt' \int_{t_{n-1}}^{t'} dt'' C(t' - t''), & n = n', \end{cases} \tag{13}$$

where $C(t)$ is the bath autocorrelation function

$$C(t) = \int_0^\infty d\omega J(\omega) \left[ \coth\left( \frac{\omega}{2T} \right) \cos(\omega t) - i \sin(\omega t) \right], \tag{14}$$

with temperature measured in units of frequency and with the spectral density $J(\omega) = \sum_i |g_i|^2 \delta(\omega_i - \omega)$.

The summand of the discretised path integral in Eq. (10) can be interpreted as the components of an $N$-index tensor $A^{j_N, j_{N-1}, \dots, j_1}$. This tensor is an ADT of the type originally proposed by Makri and Makarov[28,29]. We will show below that this $N$-index tensor can also be written as tensor network consisting of $N(N+1)/2$ tensors with, at most, four legs each and that this network can be contracted using standard MPS-MPO contraction algorithms[35,36]. First we gather terms in the inner piece of the double product in Eq. (10) into a single object, which we write as components of an $n$-index tensor

$$\mathcal{B}^{j_n, j_{n-1}, \dots, j_1} = \prod_{k=0}^{n-1} I_k(j_n, j_{n-k}). \tag{15}$$

Next, we define the $(2n - 1)$-index tensors

$$B^{j_n, j_{n-1}, \dots, j_1}_{i_{n-1}, \dots, i_1} = \left( \prod_{k=1}^{n-1} \delta^{j_{n-k}}_{i_{n-k}} \right) \mathcal{B}^{j_n, j_{n-1}, \dots, j_1}, \tag{16}$$

for $n > 1$, and the 1-index initial ADT:

$$A^{j_1} = \mathcal{B}^{j_1} \rho^{j_1}(\Delta). \tag{17}$$

We may now evolve this ADT in time iteratively by successive contraction of tensors. This process is shown graphically in Fig. 1c. The first contraction produces a 2-index ADT which describes the full state and history at the second time point:

$$A^{j_2, j_1} = B^{j_2, j_1}_{i_1} A^{i_1}. \tag{18}$$

We next contract with $B^{j_3,j_2,j_1}_{i_3,i_1}$ to produce a 3-index ADT and so on. The $n$th step of this process then looks like

$$A^{j_n, j_{n-1}, \cdots j_1} = B^{j_n, j_{n-1}, \cdots j_1}_{i_{n-1}, \cdots, i_1} A^{i_{n-1}, i_{n-2}, \cdots, i_1}, \qquad (19)$$

and the density operator for the open system at time $t_n = n\Delta$ is recovered by summing over all but the $j_n$ leg,

$$\rho^{j_n}(t_n) = \sum_{j_{n-1}, \cdots, j_1} A^{j_n, j_{n-1}, \cdots j_1}, \qquad (20)$$

from which observables can be calculated. At each iteration the size of the ADT grows by one index, since up to now we have made no cutoff for the bath memory time: we are in the 'grow' phase depicted in Fig. 1c. To compress the state after each application of this $B$ tensor, we sweep along the resulting ADT performing SVD's and truncating at each bond, throwing away the components corresponding to singular values smaller than our cutoff $\lambda_c$. This gives an MPS representation of the ADT, as given in Eq. (3). As discussed in Ref.[57], we must in fact sweep both left to right and then right to left to ensure the most efficient MPS representation is found. If no bath memory cutoff is made, this whole process is repeated until the final time point is reached at $n = N$.

The $(2n-1)$-index propagation tensor, $B$, can be represented as an MPO such that the above process of iteratively contracting tensors becomes amenable to standard MPS compression algorithms[35,36]. The form required is

$$B^{j_n, j_{n-1}, \cdots, j_1}_{i_{n-1}, \cdots, i_1} = [b_0]^{j_n}_{\alpha_1} \left( \prod_{k=1}^{n-2} [b_k]^{\alpha_k, j_{n-k}}_{\alpha_{k+1}, i_{n-k}} \right) [b_{n-1}]^{\alpha_{n-1} j_1}_{i_1}, \qquad (21)$$

where we define the rank-4 tensor

$$[b_k]^{\alpha, j}_{\alpha', i} = \delta^{\alpha}_{\alpha'} \delta^j_i I_k(\alpha, j), \qquad (22)$$

and the rank-2 and rank-3 tensors appearing at the ends of the product are

$$[b_0]^j_{\alpha'} = \delta^i_{\alpha} [b_0]^{\alpha, j}_{\alpha', i} = \delta^j_{\alpha'} I_0(j, j), \qquad (23)$$

and

$$[b_{n-1}]^{\alpha, j}_i = \sum_{\alpha'} [b_{n-1}]^{\alpha, j}_{\alpha', i} = \delta^j_i I_{n-1}(\alpha, j). \qquad (24)$$

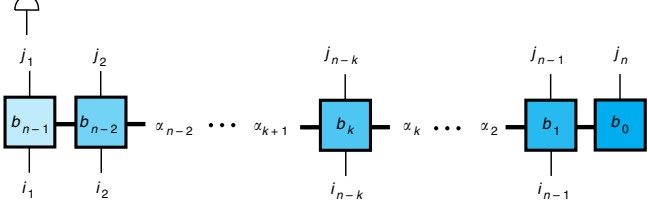

**Fig. 5** Tensor network diagram depicting the MPO decomposition of the rank-$(2n+1)$ tensor, $B$. The squares show the $b_k$ tensors in Eqs. (22)–(24), with $k$ increasing right to left. The $i_n$ and $j_n$ tensor indices correspond to the vertical legs with $n$ increasing from left to right. When $n = K$ the $j_1$ leg is summed over to give the rank-$2K$ propagation phase MPO, represented in the figure by contraction with a rank-1 object; the $d^2$-dimensional vector whose elements are all equal to one

Upon substituting these forms, Eqs. (22)–(24), into Eq. (21) it is straightforward to verify that we recover the expression Eq. (16). The rank-$(2n-1)$ MPO, $B^{j_n, j_{n-1}, \cdots, j_1}_{i_{n-1}, \cdots, i_1}$, is represented by the tensor network diagram in Fig. 5.

We note it has recently been shown that if the spectrum of $O$ has degeneracies, then part of the sum in Eq. (10) can be performed analytically, vastly reducing computational cost of the ADT method for systems where the environment only couples to a small subsystem[58]. Here we can further exploit the fact that, even when there is no degeneracy in the $d$ eigenvalues of $O$, there is always degeneracy in the $d^2$ differences between its eigenvalues, $O^-_j$, that is, $d$ of these differences are always zero. Using the same partial summing technique described in Ref. [57] we can thus reduce the dimension of the internal indices of the rank-$(2n-1)$ MPO, Eq. (21), from $d^2$ to $d^2 - d + 1$. Furthermore, if the eigenvalues of $O$ are non-degenerate but evenly spaced, as is the case for spin operators, then there are only $2d - 1$ unique values of $O^-_j$, allowing us to reduce the size of the $b_k$ tensors, Eq. (22), from $\mathcal{O}(d^8)$ to $\mathcal{O}(d^6)$.

The finite memory approximation can now be introduced by throwing away information in the ADT for times longer than $\tau_c = K\Delta$ into the system's history. To do this we write

$$[b_k]^{\alpha, j}_{\alpha', i} = \delta^{\alpha}_{\alpha'} \delta^j_i \qquad k > K. \qquad (25)$$

Thus, when propagating $A^{j_n, \cdots, j_1}$ beyond the $K$th timestep only indices $j_n$ to $j_{n-K+1}$ have any relevance and we can sum over the rest. The way we do this in practice is to define the $2K$-leg tensor MPO

$$B^{j_{K+1} \cdots, j_2}_{i_K, \cdots, i_1} = \sum_{j_1} B^{j_{K+1}, j_K, \cdots, j_1}_{i_K, \cdots, i_1}, \qquad (26)$$

such that contraction with a rank-$K$ MPS is equivalent to first growing the MPS by one leg and then summing over (i.e. removing) the leg which is earliest in time. Repeating this contraction propagates an $A$-tensor MPS forward in time, but maintains its rank of $K$ for all timesteps $n > K$. This is what we show in the 'propagate' phase of Fig. 1c. For some spectral densities, it is possible to improve the convergence with $\tau_c$ by making a softer cutoff[59,60], but since TEMPO can go to very large values of $K$ this is not necessary here.

For time-independent problems (as we study here), the 'propagate' phase involves repeated contraction with the same MPO, Eq. (26), which is independent of the timestep. To make this clear, it is convenient to change our index labelling (which, so far has referred to the absolute number of timesteps from $t = 0$). We will instead relabel the indices on the MPO and MPS as follows: $B^{j_{K+1}, \cdots, j_2}_{i_K, \cdots, i_1} \rightarrow B^{j_1, \cdots, j_K}_{i_1, \cdots, i_K}$ and $A^{j_n, \cdots, j_{n-K+1}} \rightarrow A^{j_1, \cdots, j_K}(t_n)$. The indices now refer to the distance back in time from the current time point. To summarise, with the new labelling we first grow the initial state into a $K$-index MPS, $A^{j_1, \cdots, j_K}(\tau_c)$, and then propagate as:

$$A^{j_1, \cdots, j_K}(t + \Delta) = B^{j_1, \cdots, j_K}_{i_1, \cdots, i_K} A^{i_1, \cdots, i_K}(t), \qquad (27)$$

and the physical density operator is found via

$$\rho^{j_1}(t) = \sum_{j_2, \cdots, j_K} A^{j_1, \cdots, j_K}(t). \qquad (28)$$

Having described the TEMPO algorithm we now briefly analyse the computational cost of applying it to the SBM of Eq. (4). In Fig. 6a we plot the total size, $N_{tot}$, of the MPS and maximum bond dimension, $\lambda_{max}$, used to obtain converged results in Fig. 2 against coupling strength with $K = 200$. We find the most computationally demanding regime to be around $\alpha = 0.5$, the point of crossover from underdamped to overdamped oscillations of $S_z$. We find the CPU time required is linear in the total memory requirement. For the largest memory required (at $\alpha = 0.5$), the time to obtain 500 data points using TEMPO on the HPC Cirrus cluster was $\approx 20.5$ h. In Fig. 2b we show how $N_{tot}$ grows with $K$ for different values of $\alpha$. For $\alpha = 0.1, 0.5$ we see quadratic growth with $K$, while for couplings near and above the phase transition, $\alpha = 1, 1.5$, the growth is only linear. Both cases

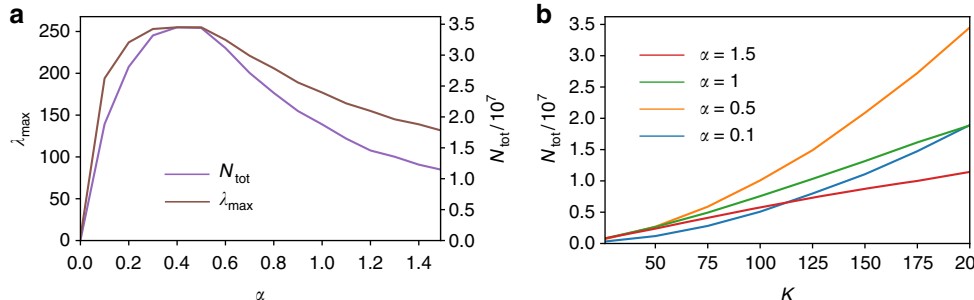

**Fig. 6** Memory requirements of the TEMPO algorithm. We show both the total size of the final MPS, $N_{tot}$, and the maximum bond dimension, $\lambda_{max}$, as a function of: **a** coupling $\alpha$ (at $K = 200$) and **b** memory cutoff $K$ (for the various values of $\alpha$ indicated)

thus represent polynomial scaling, a substantial improvement on the exponential scaling of the standard ADT method for which one has $N_{tot} = 4^K$.

**Mapping two spins in a common environment to a single spin model.** We show here how to map Eq. (6) describing a pair of spin-1/2 particles in a common environment onto Eq. (4), a single spin-1/2 SBM. The Hamiltonian Eq. (6) has the property that the total $z$-component of the two-spin system is conserved, $[S_{z,a} + S_{z,b}, H] = 0$. Thus, the problem can be separated into three distinct subspaces: the two states with the spins anti-aligned ($S_{z,a} + S_{z,b} = 0$) form one subspace and the two aligned spin states ($S_{z,a} + S_{z,b} = \pm 1$) are the other two. The one-dimensional subspaces with aligned spins cannot evolve in time; hence, all non-trivial dynamics in this model happen in the $S_{z,a} + S_{z,b} = 0$ subspace. We therefore focus on this subspace. By doing so, we may subtract a term proportional to $S_{z,a} + S_{z,b}$ from the system–bath coupling in Eq. (6). The remaining system–bath interaction is given by

$$\frac{1}{2}\left(S_{z,a} - S_{z,b}\right)\sum_i\left(|\tilde{g}_i|a_i + |\tilde{g}_i|a_i^\dagger\right). \quad (29)$$

The effective coupling here is $|\tilde{g}_i| = |g_{i,a} - g_{i,b}| = 2g_i\sin[\mathbf{k}_i \cdot (\mathbf{r}_a - \mathbf{r}_b)/2]$. These couplings lead to a modified effective spectral density[13,61],

$$J(\omega) = 2J_p(\omega)(1 - F_D(\omega R)), \quad (30)$$

where $J_p(\omega)$ is the actual density of states of the bath. The function $F_D(\omega R)$ arises from angular averaging in $D$-dimensional space, and so crucially depends on the dimensionality of the environment. Specifically we have:

$$F_D(x) = \begin{cases} \cos(x), & D = 1, \\ J_0(x), & D = 2, \\ \mathrm{sinc}(x), & D = 3, \end{cases} \quad (31)$$

where $J_0(x)$ is a Bessel function. We note that $F_D(\omega R) \to 0$ as $R \to \infty$ for $D > 1$, due to the diminishing effect of the environment-induced coupling in higher dimensions. (When considering $R \to \infty$, we should note that in the original Hamiltonian we neglected any retardation in the Heisenberg interaction.) At small separations, $R \to 0$, $F_D(\omega R) \to 1$ and so $J(\omega) \to 0$ for all $D$ due to the loss of relative phase shift between the couplings of the anti-aligned states to the environment.

For the bare density of states $J_p(\omega)$, we consider a simple model of e.g. a quantum dot in a phonon environment, for which the coupling constants appearing in the Hamiltonian, Eq. (6), have $g_i \sim \sqrt{\omega_i}$[19]. This means that in the continuum limit the spectral density for a $D$-dimensional environment is

$$J_p(\omega) = \frac{\alpha}{2}\frac{\omega^D}{\omega_c^{D-1}}e^{-\omega/\omega_c}, \quad (32)$$

where $\omega_c$ describes a high-frequency cutoff and $\alpha$ is the strength of the interaction with the environment.

**Data and code availability.** The datasets generated during and/or analysed during the current study are available at: https://doi.org/10.17630/44616048-eaac-4971-bbff-1d36e2cef256. The TEMPO code is available at https://doi.org/10.5281/zenodo.1322407.

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

## Acknowledgements

We thank T.M. Stace for useful discussions and J. Iles-Smith for comments on an earlier version of this paper. A.S. acknowledges a studentship from EPSRC (EP/L505079/1). P.K. acknowledges support from EPSRC (EP/M010910/1). D.K. acknowledges support from the EPSRC CM-CDT (EP/L015110/1). J.K. acknowledges support from EPSRC programs 'TOPNES' (EP/I031014/1) and 'Hybrid Polaritonics' (EP/M025330/1). B.W.L. acknowledges support from EPSRC (EP/K025562/1). This work used EPCC's Cirrus HPC Service (https://www.epcc.ed.ac.uk/cirrus).

## Author contributions

The TEMPO code was developed by A.S., P.K. and D.K., following the identification of the MPS representation by J.K. Analysis of the two applications was performed by A.S., P.K. and B.W.L.. The project was directed by J.K. and B.W.L. All authors contributed to the writing of the manuscript.

## Additional information

**Competing interests:** The authors declare no competing interests.

