## [Peer Review File · Nature Communications]

REVIEWERS' COMMENTS:

Reviewer #2 (Remarks to the Author):

The authors of the revised article have attempted to argue the case that the article is both innovative and relevant to the current journal. The original disagreement hinges on whether or not recognizing that the quasi-adiabatic path integral contains quantities that can be compressed as a matrix product state is, in as of itself, an original observation. Given that tensor network compression techniques have been applied to a wide variety of high dimensional problems, by the author's own admission - including to real-time path integrals, of which the current example is a special case - I still contend that this is not a major conceptual step. But it is equally true that this *particular* step has not previously been written down. (Although note that there are related alternatives the authors do not seem to be aware of: see for example, arxiv.org/abs/1801.07418 which defines "reservoir tensor networks" that describe open quantum system dynamics for finite time memory kernels, closely related to the factorization of the auxiliary density operators in this paper, not to mention the use of tensor techniques (such as multi-layer MCTDH) to simulate long time dynamics of open quantum system with explicit baths).

I think we probably cannot reach agreement on the issue of originality. But regardless of whether this particular step is original or not, the work is nice and of high quality. The remaining issue for publication in my mind would seem to be whether or not this advance is of general interest.

I suggest that the decision be made on the following grounds:

(i) Is this journal one that typically publishes advances in open quantum system dynamics? My observation is that other advances in such methods, e.g. the QUAPI method itself, hierarchical equations of motion, inchworm (e.g. the methods mentioned by the authors), not to mention the foundational papers of tensor network dynamics algorithms, such as iTEBD, have all been published in other journals;

(ii) Are the applications of the technique sufficiently interesting and novel that they merit publication in a general interest journal? We have agreed that the spin-boson results do not; the two-spin case, is interesting (given that the bath has long time correlations) in that it demonstrates the strengths of TEMPO method - although when the paper states that other methods are not available, I note that no numerical calculations are done to show that other methods could not achieve the same result. But other than proving the strength of the method, I do not know that it is interesting. As an analogy, imagine that one presented a calculation on the Helium atom which obtained the non-relativistic ground-state energy to 100 d.p., instead of the ~ 30 d.p. to which it is currently known. That would be a significant technical achievement, but is it of general interest?

Thus it remains my conclusion that while the authors clearly would like to publish this paper in this journal, it does not seem like a paper for the audience that reads Nature communications, which I think of as containing papers that can be read by an experimentalist as well as a theorist. It seems to me the kind of paper that should really be in a Physical Review journal.

Reply to Reviewer's Comments

1 Response to Second Report of Referee # 2

We respond to the comments as appropriate below (our response in black, the referee comments in red).

The authors of the revised article have attempted to argue the case that the article is both innovative and relevant to the current journal. The original disagreement hinges on whether or not recognizing that the quasi-adiabatic path integral contains quantities that can be compressed as a matrix product state is, in as of itself, an original observation. Given that tensor network compression techniques have been applied to a wide variety of high dimensional problems, by the author's own admission - including to real-time path integrals, of which the current example is a special case - I still contend that this is not a major conceptual step. But it is equally true that this *particular* step has not previously been written down. (Although note that there are related alternatives the authors do not seem to be aware of: see for example, arxiv.org/abs/1801.07418 which defines "reservoir tensor networks" that describe open quantum system dynamics for finite time memory kernels, closely related to the factorization of the auxiliary density operators in this paper, not to mention the use of tensor techniques (such as multi-layer MCTDH) to simulate long time dynamics of open quantum system with explicit baths).

The paper noted by the referee (arxiv.org/abs/1801.07418) does describe a possible method for performing open system simulations using a reservoir tensor networks, but it does not include any implementation of this idea - and so it cannot be subject to efficiency tests on standard models.

I think we probably cannot reach agreement on the issue of originality. But regardless of whether this particular step is original or not, the work is nice and of high quality. The remaining issue for publication in my mind would seem to be whether or not this advance is of general interest.

I suggest that the decision be made on the following grounds:

(i) Is this journal one that typically publishes advances in open quantum system dynamics? My observation is that other advances in such methods, e.g. the QUAPI method itself, hierarchical equations of motion, inchworm (e.g. the methods mentioned by the authors), not to mention the foundational papers of tensor network dynamics algorithms, such as iTEBD, have all been published in other journals;

(ii) Are the applications of the technique sufficiently interesting and novel that they merit publication in a general interest journal? We have agreed that the spin-boson results do not; the two-spin case, is interesting (given that the bath has long time correlations) in that it demonstrates the strengths of TEMPO method - although when the paper states that other methods are not available, I note that no numerical calculations are done to show that other methods could not achieve the same result. But other than proving the strength of the method, I do not know that it is interesting. As an analogy, imagine that one presented a calculation on the Helium atom which obtained the non-relativistic ground-state energy to 100 d.p., instead of the 30 d.p. to which it is currently known. That would be a significant technical achievement, but is it of general interest?

Thus it remains my conclusion that while the authors clearly would like to publish this paper in this journal, it does not seem like a paper for the audience that reads Nature communications, which I think of as containing papers that can be read by an experimentalist as well as a theorist. It seems to me the kind of paper that should really be in a Physical Review journal.